# Identifying Function Determining Residues in Neuroimmune Semaphorin 4A

**DOI:** 10.3390/ijms23063024

**Published:** 2022-03-11

**Authors:** Svetlana P. Chapoval, Mariah Lee, Aaron Lemmer, Oluwaseyi Ajayi, Xiulan Qi, Andrew F. Neuwald, Achsah D. Keegan

**Affiliations:** 1Department of Microbiology and Immunology, University of Maryland School of Medicine, 655 W Baltimore St., Baltimore, MD 21201, USA; schapoval@som.umaryland.edu; 2Center for Vascular and Inflammatory Diseases, University of Maryland School of Medicine, 800 W Baltimore St., Baltimore, MD 21201, USA; mariah.lee@newberry.edu (M.L.); aaronlemmer3@yahoo.com (A.L.); oajayi17@terpmail.umd.edu (O.A.); xqi@som.umaryland.edu (X.Q.); 3Program in Oncology, Greenebaum Comprehensive Cancer Center, University of Maryland School of Medicine, 22 S Greene St., Baltimore, MD 21201, USA; 4SemaPlex LLC, Ellicott City, MD 21041, USA; 5Department of Biochemistry and Molecular Biology, University of Maryland School of Medicine, 108 N Greene St., Baltimore, MD 21201, USA; 6Institute for Genome Sciences, University of Maryland School of Medicine, 655 W Baltimore St., Baltimore, MD 21201, USA; 7Maryland Health Care System, Baltimore VA Medical Center, 10 N Greene St., Baltimore, MD 21201, USA

**Keywords:** Semaphorin 4A, Plexin B1, human Treg cells, mutated proteins, immunotherapeutics for asthma

## Abstract

Semaphorin 4A (Sema4A) exerts a stabilizing effect on human Treg cells in PBMC and CD4+ T cell cultures by engaging Plexin B1. Sema4A deficient mice display enhanced allergic airway inflammation accompanied by fewer Treg cells, while Sema4D deficient mice displayed reduced inflammation and increased Treg cell numbers even though both Sema4 subfamily members engage Plexin B1. The main objectives of this study were: 1. To compare the in vitro effects of Sema4A and Sema4D proteins on human Treg cells; and 2. To identify function-determining residues in Sema4A critical for binding to Plexin B1 based on Sema4D homology modeling. We report here that Sema4A and Sema4D display opposite effects on human Treg cells in in vitro PBMC cultures; Sema4D inhibited the CD4+CD25+Foxp3+ cell numbers and CD25/Foxp3 expression. Sema4A and Sema4D competitively bind to Plexin B1 in vitro and hence may be doing so in vivo as well. Bayesian Partitioning with Pattern Selection (BPPS) partitioned 4505 Sema domains from diverse organisms into subgroups based on distinguishing sequence patterns that are likely responsible for functional differences. BPPS groups Sema3 and Sema4 into one family and further separates Sema4A and Sema4D into distinct subfamilies. Residues distinctive of the Sema3,4 family and of Sema4A (and by homology of Sema4D) tend to cluster around the Plexin B1 binding site. This suggests that the residues both common to and distinctive of Sema4A and Sema4D may mediate binding to Plexin B1, with subfamily residues mediating functional specificity. We mutated the Sema4A-specific residues M198 and F223 to alanine; notably, F223 in Sema4A corresponds to alanine in Sema4D. Mutant proteins were assayed for Plexin B1-binding and Treg stimulation activities. The F223A mutant was unable to stimulate Treg stability in in vitro PBMC cultures despite binding Plexin B1 with an affinity similar to the WT protein. This research is a first step in generating potent mutant Sema4A molecules with stimulatory function for Treg cells with a view to designing immunotherapeutics for asthma.

## 1. Introduction

Members of the semaphorin family of secreted and transmembrane proteins were initially characterized as axon-guidance molecules critical for neuronal development [1]. They function as monomers, homodimers, or heterodimers [2] to suppress or promote, for example, axonal outgrowth (reviewed in ref. [1]), angiogenesis [3,4,5,6], cancer [3,7,8,9], and inflammation [10]. In the immune system, several semaphorins act as immune checkpoints (reviewed in ref. [11,12]). All vertebrate semaphorins functionally use plexins as receptors in non-lymphoid [13,14] and lymphoid [15,16,17] tissues. We previously have shown that two Class IV neuroimmune semaphorin molecules, Sema4A and Sema4D, display the opposite effects in vivo in allergic airway inflammation. Sema4A displayed a suppressive role and Sema4D exhibited an inflammation-potentiating function [18,19]. We recently reported that Sema4A stabilized human peripheral blood Treg cells and increased CD25 and Foxp3 expression by these cells [17]. Moreover, it induced the formation of new Treg cells in in vitro PBMC cultures [17]. Treg cells present a special interest for potential cellular therapy of allergic asthma as they directly and specifically suppress effector Th2 cells, thus lowering or even abrogating the allergen sensitization-induced tissue effects [20,21,22]. Recent single cell transcriptomic studies defined the molecular properties of Th and Treg cells in asthma and provided further detailed molecular proof that a Th/Treg imbalance in response to allergen plays a critical role in disease pathogenesis [22]. The identification of molecules which regulate Treg cell stability is critically important for Treg-mediated asthma therapy as a proallergic inflammatory environment, and can modify suppressor Treg cells toward their pathogenic phenotype and function [21].

Sema4A has seven reported receptors [3,11,23,24,25]. Although there is no detailed structural information on Sema4A or its interaction with the counter receptors, there is ample information on other members of the semaphorin family. The Sema domain is a distinctive structural and functional domain of the semaphorin superfamily which includes semaphorins, plexins, and receptor tyrosine kinases (RTKs) MET and recepteur d’origine nantais (RON) members [26,27,28]. Kolodkin and associates in 1993 first defined the sema domain in the analysis of DNA sequences, encoding several insect, vertebrate, and viral semaphorins; they showed that this domain is located in the N-terminus of semaphorin molecules and is approximately 500 aa long [27]. The Sema domain has a characteristic set of cysteine residues that form four disulfide bonds to stabilize the structure. Crystal structures of Sema3A, Sema4D, and MET have shown that the Sema domain has a seven bladed β propeller topology [29,30,31]. Each blade contains a four-stranded (strands A to D) antiparallel beta-sheet. The propeller-like appearance comes from the circular arrangement of beta-sheets where the inner strand of each sheet runs in parallel to a central axis, whereas the outer strand runs perpendicular [26]. The inner strand of each blade (A) lines the channel at the center of the propeller, with strands B and C of the same repeat radiating outward, and strand D of the next repeat forming the outer edge of the blade. The Sema domain uses a ‘loop and hook’ system to close the circle between the first and the last blades. The blades are constructed sequentially, with an N-terminal β-strand closing the circle by providing the outermost strand (D) of the seventh (C-terminal) blade. The β-propeller is further stabilized by an extension of the N-terminus, providing an additional, fifth β-strand on the outer edge of blade 6 [26,27,28,29,30,31].

Since there is no structural information specific for Sema4A, to predict residues critical for Treg stabilizing activity via Plexin B1, we performed Bayesian Partitioning with Pattern Selection (BPPS) and Structurally Interacting Pattern Residues’ Inferred Significance (SIPRIS) analyses to identify residues likely responsible for functional specialization and, in particular, for protein–protein interactions. BPPS [32,33] identifies residues that best distinguish each protein subgroup from other closely related subgroups, which are therefore likely responsible for each subgroup’s specific function. SIPRIS [34] identifies statistically significant structural interactions involving a BPPS-defined set of residues.

Using this approach, we identified amino acid residues in human Sema4A that we predicted would mediate Sema4-specific responses (M198 and F223). We found that the F223A mutant Sema4A protein demonstrated a similar affinity to Plexin B1 but a lack of stimulatory activity for Treg cells when compared to the WT protein. This result will be informative for the future design of therapeutics for allergic asthma, given that WT Sema4A functions as a downregulatory molecule for allergen-induced disease in mice [10,11,18].

## 2. Results

### 2.1. Sema4A and Sema4D Elicit Opposite Effects on Treg Cells in PBMC Cultures

Both Sema4A and Sema4D can functionally engage Plexin B1 [10,11,35,36]. We previously showed that Sema4A mediates human peripheral Treg cell stability utilizing Plexin B1 [17]. To compare the effects of exogenous rhSema4D and rhSema4A on human peripheral blood Treg cells, we established in vitro cultures of human PBMCs as described previously [17]. The cells were harvested 48 h after the set-up, stained with fluorochrome-labelled Ab to corresponding markers, and analyzed by FACS. The gating strategy for FACS analysis is shown in Appendix A. We observed that similarly to rhSema4A, exposure of PBMC to rhSema4D did not affect the total number of CD4+ T cells in cultures (Figure 1A,C, left panel). However, rhSema4D decreased the relative number of CD4+ Foxp3+ Treg cells to 2.5 ± 0.3% compared to 3.6 ± 0.5% with medium alone, *p* = 0.040 (Figure 1B,C, right panel), whereas Sema4A increased Tregs to 5.5 ± 0.4%, *p* = 0.012 vs. medium control, *p* = 0.001 vs. Sema4D. There was evidence that rhSema4D induced some co-stimulatory activity, since we observed an increase in CD4+ CD25^high^ T cell number to 7.4 ± 0.4% from 4.9 ± 0.3% in cultures with medium alone (Figure 1 C, middle panel, *p* < 0.012). This is consistent with previous reports that Sema4D acts as a positive costimulatory molecule for CD4+ T cells [11,19,25]. Therefore, these individual members of Class IV semaphorin family display the opposite effects on human Treg cells in vitro.

### 2.2. Sema4A Exhibits a Higher Binding Affinity to Plexin B1 Than Sema4D

To determine whether the opposite effect of Sema4A versus Sema4D on Treg cells in PBMC cultures was a result of distinct receptor-binding affinities, we next examined the affinity of rhSema4A and rhSema4D for Plexin B1 binding utilizing LRA ELISA assay. When Plexin B1 was immobilized on a microplate and increasing concentrations of soluble rhSema4 were applied to the wells, we detected the prominent binding of both Sema4 molecules to their common Plexin B1 receptor (Figure 2, upper and lower left panels). The equilibrium dissociation constant for Sema4A was 19.9 ± 5.7 nM (Figure 2, upper left panel) and for Sema4D it was similar at 23.5 ± 7.3 nM (Figure 2, lower left panel). When Sema4A was immobilized, the addition of rhPlexin B1 in solution resulted in a dissociation constant of 35.4 ± 6.2 nM (Figure 2, upper right panel). However, immobilization of Sema4D to the plate greatly reduced its interaction with soluble Plexin B1 (Figure 2, lower right panel). The corresponding Scatchard plots from one out of three ELISA experiments are shown as inserts in the panels of Figure 2. Altogether, these results confirm the Sema4-Plexin B1 interaction and provide the corresponding Kd values. We also examined the affinity of Sema4A for NRP1. We observed that Sema4A binds NRP1 alone with high affinity with an equilibrium dissociation constant of 28.27 nM (Appendix A). While Sema4A binding to NRP1 has been demonstrated in LRA ELISA previously, the dissociation constant was not calculated in that study [37].

### 2.3. Sema4D and Sema4A Compete In Vitro for Plexin B1 Binding

We next investigated the competition between Sema4A and Sema4D for the Plexin B1 receptor. For that purpose, His-tagged Plexin B1 was coupled to HIS-Select nickel-coupled microplate and increasing concentrations of rhSema4D were applied to wells, whereas in separate wells the increasing concentrations of rhSema4A were added to the microplate wells containing 2 μg/mL of Sema4D (Figure 3). rhSema4D binds plate-immobilized Plexin B1 with high affinity (Figure 3A), whereas the addition of rhSema4A inhibits such binding by >70% (Figure 3B). These results demonstrate that Sema4D and Sema4A compete in vitro for Plexin B1. Therefore, they could be competing for Plexin B1 ligands in vivo, as one ligand–receptor interaction could mask another ligand’s binding domain.

### 2.4. BPPS Analysis of Sema Domains

To explore the basis for the differential effects of Sema4A and Sema4D, we performed computational analyses. First, we performed a BLAST sequence alignment of human Sema4A and Sema4D proteins demonstrating a 40% homology between these two Class IV semaphorin sequences (Appendix A). Next, we utilized BPPS to generate a Sema-domain hierarchy as shown in Figure 4 and annotated in Appendix A. BPPS generated contrast alignments showing the Sema residues distinctive of subgroups along the lineage: Sema-domain superfamily (root), Sema3,4 family (node 2), and Sema4A subfamily (node 14) (Appendix A).

The advantage of this analysis over a conventional phylogenetic tree [26] is that it automatically clusters a large number of protein sequences into functionally related subgroups while concurrently defining subgroup-specific conserved residues [32,38]. Because such residues are conserved across divergent Sema domains and thus correspond to persistent constraints, they are likely to be biologically relevant. BPPS classifies the Sema domains into five families, four of which were subdivided further into 12 subfamilies, with one of these again further subdivided into three sub-subfamilies and with two of these giving rise to one sub-subfamily each. Appendix A gives the names of the Sema domains assigned to each node in Figure 4. The BPPS-generated contrast alignments for Sema4A in Appendix A correspond to the lineage leading from the Sema superfamily (i.e., the root) to Sema4A (i.e., node 14) within the hierarchy. Hence, Appendix A highlights those residues likely determining the functional specificity of Sema3, 4 proteins, and of Sema4A. For example, the bottom alignment highlights the aligned residues that most distinguish Sema4A domains (termed the foreground) from other Sema3,4 domains (termed the background). Note that the pattern residues were defined in the absence of structural information, and, conversely, that the structural clusters in the following section were defined in the absence of sequence information.

### 2.5. SIPRIS Analysis of Sema Domain

We used SIPRIS to determine whether certain BPPS-defined residues are correlated with structurally-defined features—such as, for example, whether certain residues are correlated with a Sema4/Plexin B1 interface. Using the BPPS pattern residues for both the Sema4D and Plexin B1 subfamilies (nodes 22 and 12, respectively, in Figure 4 and Appendix A), SIPRIS identified both Sema4D residues implicated in Plexin B1 binding (*p* = 2 × 10^−5^), and Plexin B1 residues implicated in Sema4D binding (*p* = 6 × 10^−7^) (Figure 5A). The Sema4D subfamily (node 22) belongs to the Sema3,4 family (node 2 in Figure 4) that also includes the Sema4A subfamily (node 14), which is of primary interest here. The residues most distinctive of the Sema3,4 family (node 2) tend to cluster on one side of the interface with plexinB1 (*p* = 5.5 × 10^−9^) (red sidechains in Figure 5B), whereas those distinctive of the Sema4D subfamily (node 22) tend to cluster on the other side of this interface (orange sidechains in Figure 5B). This suggests a potential mechanism for specific activation of Plexin B1 by Sema4D (as opposed to other Sema3,4 subfamilies). Notably, a few of the Sema4D residues form a second cluster at the Sema4D homodimeric interface (circled in Figure 5B). Using these same BPPS residue sets and structural coordinates, we performed Sema4D and Plexin B1 BPPS-SIPRIS-core-clustering. This involves constructing a set of the sequentially-generated structural clusters (without utilizing information regarding the Sema4D–Plexin B1 interface or the BPPS analyses) and then reporting, for each BPPS category, the most significantly correlated structural cluster. Significant clusters were found for both Sema4D and Plexin B1 (*p* = 7 × 10^−6^ and *p* = 3 × 10^−8^, respectively) (Figure 5C). Each of these clusters contain the corresponding BPPS-structural cluster defined by the Sema4D–Plexin B1 interface in Figure 5A. A structural analysis suggested that other proteins may bind to the region adjacent to the Sema4D–Plexin B1 interface (Figure 5D). As an example, we modeled the hypothetical binding of the β-chain of Macrophage Stimulating Protein (MSP) to Sema4D (Figure 5D).

### 2.6. Sema4A Analysis

Although BPPS identified residues distinctive of the Sema4 subfamily, the lack of a Sema4A structure prohibits the sort of SIPRIS analysis performed for Sema4D, as just described. Nevertheless, based on homology (Appendix A), we have mapped to the Sema4D–Plexin B1 structure seven of the most distinctive Sema4-subfamily residues that also map to the Plexin B1 interface (residues with lime-colored sidechains in Figure 6). The residues M198, N199, F223, L357, K359, P383, and S384 (see Appendix A) were identified as targets for mutagenesis, with M198 and F223 having the highest priority given their structural locations and the strength of the Sema4A-specific constraint imposed upon them. Interestingly, P383 and S384 are located close to a residue previously predicted to interact with Plexin B1 (V381) [39]. Four of these seven residues are within a region (from K352 to M398) that is highly variable among various Sema domain proteins, suggesting an important role for this region in Sema-domain functional specificity. Other residues distinctive of the Sema4A within this highly variable region (such as K353, S362, G382, and F396) could also serve as the potential targets for mutagenesis, however, they are not as differentially conserved as those mentioned above. The residues F257 and Q329 are predicted to be near a putative Plexin B1 binding site and near the homodimeric interface.

### 2.7. Effect of Mutations in Sema4A on Binding to Plexin B1 and Treg Cell Stabilizing Function

Sema4A proteins with either M198 or F223 mutated to alanine (denoted as M198A and F223A, respectively) (Figure 6 and Appendix A) were tested in vitro for Plexin B1 binding in the LRA ELISA employing HIS-Select nickel-coupled microplates. We found that both mutants, when used in solution, demonstrated binding affinities to the plate-bound Plexin B1 comparable to that observed for WT Sema4A (Figure 7A), with high equilibrium dissociation constants of 2.55 ± 1.48 nM for WT, 4.19 ± 0.09 for M198A, and 2.82 ± 0.75 nM for F223A. When the Sema4A proteins were coated directly on Immulon 2 plates with increasing concentrations of rhPlexin B1 applied in solution, the difference in binding between the two mutant proteins was as follows: 2.5 nM for WT Sema4A, 1.7 nM for M198A mutant, and 3.5 nM for F223A mutant proteins with corresponding Kd ranges (0.84–5.61), (−0.25–7.63), and (1.64–6.60) (Figure 7B). Thus, the Sema4A-M198A and Sema4A-F223A mutants can bind Plexin B1.

The mutant proteins were tested for their stimulatory effect on Treg cells in PBMC cultures, as described previously [17]. As expected, WT Sema4A displayed a highly specific effect on Treg cells, as it increased the relative number of CD4+Foxp3+ cells among PBMCs from 3.96 ± 0.46% in cultures with medium only to 6.18 ± 0.58% in cultures with 100 ng of WT Sema4A (Figure 8A,C). At the same time, the M198A Sema4A mutant demonstrated a more modest effect on the relative number of Tregs to 6.07 ± 0.61% of CD4+Foxp3+ cells in cultures (*p* < 0.008 vs. medium control). In contrast, the F223A mutant had little to no Treg-stabilizing activity when compared to the media control (4.35 ± 0.52% vs. 3.96 ± 0.46%, respectively) and was substantially less effective than WT Sema4A (4.35 ± 0.52% vs. 6.18 ± 0.58%, respectively, *p* < 0.038) (Figure 8A,C). Next, we compared the stimulatory effects of an increased dose of 1 μg/mL for all selected proteins on high CD25 expression on T cells (Figure 8B,C). In the PBMC cultures with medium, only 3.83 ± 0.5% of CD4+ T cells expressed high levels of CD25, while in the presence of WT Sema4A this increased in a dose-dependent manner (Figure 8C). These CD25^high^ T cells co-expressed Foxp3 (Figure 8B). Notably, the CD25^high^ expression on Treg cells for WT Sema4A was twice that of the F223A mutant. Therefore, a mutation of phenylalanine at 223 position of Sema4A led to a significant disfunction of this protein in terms of its stimulatory effect on human Treg cells.

## 3. Discussion

We show here that two members of the Class IV semaphorin family display opposite effects on human Treg cells in PBMC cultures. While Sema4A displays a Treg stabilization effect by both increasing CD25 and Foxp3 expression on Treg cells and increasing the number of Treg cells in the in vitro cultures of PBMC, Sema4D decreases those parameters. These findings agree with our previously published in vivo data, where we reported that the lack of Sema4D in mice led to a lower allergic inflammatory response to OVA challenges accompanied by a higher number of Treg cells, whereas the lack of Sema4A increased allergic responses associated with a decreased Treg cell number [18,19]. In Sema4D^-/-^ mice we observed a downregulation of airway eosinophilic infiltration, mucus secretion, and lower anti-OVA IgE levels as compared to similarly treated WT mice. This could be related, at least in part, to a suppressive effect of Sema4D on Treg cells which are well-known for their downregulatory effects on Th2 effector cells and for controlling allergic diseases [20,21,22,40]. We reported previously that the overall percentage of CD4 + CD25+ T cells did not vary between the PBMC cultures with or without rhSema4A, however, the relative number of CD4 + CD25^high^Foxp3^high^ Treg cells was significantly potentiated, almost doubling in numbers [17]. These Treg cells were found to stain positively for the Sema4A receptor Plexin B1 [17]. Of note, these human Tregs did not express NRP1 [17].

The inhibitory effect of the external Sema4D on Treg cells in vitro was reported recently by Xie and associates [41]. This study used PBMC isolated from blood samples of normal control donors and of patients with ankylosing spondylitis. The external soluble Sema4D led to CD4+ T cell proliferation and Th17 cell differentiation. However, the expression of Foxp3 was significantly downregulated in CD4+ T cells obtained from both groups, whereas the expression of RORγπ transcription factor critical for Th17 cell differentiation was upregulated. The authors reported that the relative number of Treg cells dropped from 3.56% to 2.03% with Sema4D exposure. The significant increases in the numbers of Th17 cells and decreases in the numbers of Treg cells were reported to be strongly associated with the pathogenesis of ankylosing spondylitis [42].

Treg cells play critical regulatory and suppressive roles in many chronic inflammatory diseases [20,21,22]. A previous study from our laboratory has demonstrated that Treg cells have an additional function in allergic asthma besides the well-established suppression of downstream effect of Th2 responses such as airway eosinophilia, lung tissue remodeling, and IL-5 production, as they also block T cell recruitment and homing into inflamed lung tissues [43]. Our work also showed that a small number of Treg cells is sufficient to abrogate asthmatic response in mice. Only 1.5% of lung-draining LN cells from STAT6xRAG2 ^−/−^ mice that received CD4 + CD25- T cells expressed Foxp3. These cell recipients developed a significant level of eosinophilic lung inflammation (38% of total broncho-alveolar lavage cells) in response to allergen treatment. In contrast, STAT6xRAG2^−/−^ mice that received CD4 + CD25+ Tregs contained a higher proportion of donor CD4 + Foxp3+ Treg cells (5.73%) in their lung-draining LNs. This Treg transfer led to a complete abrogation of eosinophilic lung inflammation.

We aimed to define the amino acid residues in Sema4A critical for Plexin B1 binding. As the Sema4A crystal structure is unavailable, we used BPPS and SIPRIS to define these residues, which also allowed us to reveal molecular distinctions among divergent Sema domain-containing subgroups such as semaphorins, plexins, MET, and RON members of RTKs superfamily. Furthermore, our BPPS and SIPRIS analyses of Sema domains, Sema4D–Plexin B1 interaction sites, and Sema4D sequence substitution to Sema4A provided clues regarding co-conserved residues among Sema domains and Sema4 subclass proteins. Moreover, these analyses allowed us to label Plexin B1-binding residues on the Sema4A ligand. These residues are localized to the Sema4A-Plexin B1 interface and may participate in the Sema4A functions in an experimental model of asthma [44,45] and in human Treg cell stability [17]. We tested the hypothesis that two of these residues would be important for Plexin B1 binding and Treg stabilizing function. Neither the F223A nor the M198A mutation substantially altered the binding of Sema4A to Plexin B1. Nevertheless, the inferred structural locations of Sema4A residues that interact with Plexin B1 provide new insights into the molecular interface associated with Sema4A specificity and its possible mechanism of Plexin B1 activation. It is possible that multiple residues contribute to binding, and thus a single amino acid mutation would not cause a substantial loss of affinity. Based on Sema-PSI of RON RTK structure and ligand-interaction, it was proposed that all Sema domain-containing proteins homodimerize in such a way that it prevents further ligand binding that, in turn, regulates their signaling activities [46].

We focus this report on Sema4A binding to Plexin B1 (Figure 2) and, presently, omitted its binding to NRP1 (Appendix A) because human peripheral blood CD4+ T cells do not express NRP1. This is related to the signaling activity of the Sema4A-Plexin B1 interaction, whereas the Sema4A-NRP1 interaction, as discussed above, might be necessary to form a ligand–receptor complex with Plexin B1 stabilization—analogous to a previously reported Sema3-NRP-Plexin A complex function [36,47]. Neuropilins appear to be expressed only in vertebrates and represent the highly conserved, single-pass transmembrane proteins [48]. More recently, neuropilins were shown to interact with VEGFR1 and VEGFR2 and, therefore, to serve as co-receptors for vascular endothelial growth factor (VEGF) [48]. Such co-receptor function of NRPs with other receptors and, therefore with multiple ligands, allow neuropilins to play multiple roles in many cellular activities such as cell adhesion, repulsion, stabilization, and survival. Structurally, it has been reported that NRP engages Sema3 proteins using a divalent engagement, with the a1 domain of NRPs binding the Sema domain of different Sema3 molecules and the b1 domain engaging the Sema3 C-terminal domain with high affinity [48]. It was originally believed that only Class III semaphorins utilize NRPs as high affinity signaling receptors. However, it has recently been demonstrated that a Sema4A-NRP1 specific interaction is required by Treg cells to inhibit anti-tumor immune responses and showed significant promise in therapeutic intervention in inflammatory colitis in mice [37]. We plan to expand this study to include other functionally critical residues in Sema4A, their combinations, and explore the functional activity of novel mutant Sema4A proteins to treat asthma and, potentially, other chronic inflammatory conditions where Treg cells play essential regulatory roles. It is important, however, to verify that such Sema4A mutants lack binding to potentially asthma-upregulating Sema4A receptors such as ILT4 and Plexin D1 [24,49].

When applied to 4505 Sema domain sequences, BPPS/SIPRIS revealed the likely sequence and structural determinants of Sema domain functional specificity. This includes seven BPPS-defined residues highly distinctive of Sema4A and, based on sequence homology to Sema4D, potential Plexin B1 binding sites on Sema4A. Based on these considerations, we selected two Sema4A residues, M198 and F223, highly likely to be functionally important, which we mutated to alanine (M198A and F223A). Notably, F223 in Sema4A corresponds to an alanine residue in Sema4D. Since Sema4D displayed a downregulatory effect and Sema4A a stimulatory effect on human Treg cells in PBMC cultures, and, since both bind to Plexin B1, we hypothesized that the F223A mutation may eliminate Sema4A’s stimulatory effect on Tregs but not its ability to bind Plexin B1. Indeed, as it can be appreciated from Figure 7, the binding affinity of WT and mutant Sema4A proteins was similar, however, their ability to stimulate Treg cells differed.

We defined the dissociation constants for Sema4A-Plexin B1 interaction which equaled 19.9 ± 5.7 nmol when Sema4A was immobilized and Plexin B1 in solution and 35.4 ± 6.2 nmol when Sema4A was in solution (Figure 2, Figure 3 and Figure 7). For comparison, the binding of Sema4A to Plexin D1 in the BIAcore measurements showed a Kd of 8.72 nM [50]. The chip-bound monomer of Plexin B1 showed that it binds the soluble Sema4D ectodomain (Sema domain) with Kd = 5.5 mM in the surface plasmon resonance equilibrium experiments [39]. However, the study by Tamagnone and associates [51] demonstrated Kd = 0.9 nM for Sema4D–Plexin B1 interaction in cell-based ELISA when COS cells transfected with Plexin B1 DNA were exposed to soluble Sema4D fused to alkaline phosphatase. Another Sema4A receptor, NRP1, was reported to be critical for the Sema4A-mediated stability and function of mouse Tregs [37] and for the maintenance of intra-tumoral Treg cells [52].

In summary, we used the BPPS/SIPRIS approach to define the critical amino acid residues on Sema4A for Plexin B1 binding (Figure 6 and Appendix A) based on the published Sema4D–Plexin B1 interaction [39]. We generated two mutated Sema4A proteins which were further tested in vitro for their receptor binding affinity and for Treg cell stimulation in cultures. We showed here that a single mutation in the receptor-binding site of Sema4A could significantly affect the Sema4A isoform functional activity. Interestingly, the key point mutation F223A in Sema4A allowed Plexin B1 binding with a high affinity similar to that of WT protein but was non-stimulatory for the peripheral human Treg cells in vitro. Another point mutation, M198A, demonstrated high affinity binding to Plexin B1, and maintained its ability to stimulate Treg cell stability in the in vitro cultures similar to that of WT Sema4A. The fact that M198 is well conserved across diverse organisms implies that it is subject to significant selective constraints, however it does not appear to be essential for the Treg stabilizing function. It remains to be determined whether M198 is critical for other Sema4A-induced activities.

## 4. Materials and Methods

### 4.1. Proteins

Soluble recombinant human (rh) Sema4A protein was purchased from either Abnova (Taipei City, Taiwan) (Glutathione Sepharose (GST)-tagged), GenScript (Piscataway, NJ, USA) (Fc-tagged), R&D Systems (Minneapolis, MN, USA) (Fc-tagged), or Acro Biosystems (Newark, DE, USA) (His-tagged). The rhSema4A consisted of the full extracellular domain spanning AA Gly 33—His 683 (Acro Biosystems), or Gly32—Fc 683 (R&D Systems), or 1—761 with GST tag at N-terminus (Abnova). The molecular weight of rhSema4A from Acro Biosystems was 72.7 kDa, Abnova—110 kDa, R&D Systems—98.5 kDa, GenScript—110 kDa. rhPlexin B1 with a 6X His tag at C-terminus and MW = 66kDa was purchased from GenScript. Mutant Sema4A proteins with amino acids methionine and phenylalanine in positions 198 and 223, correspondingly, changed to alanine (M198A and F223A) and were prepared by GenScript. The mutant proteins were also Fc-tagged and demonstrated similar molecular sizes as the WT Sema4A-Fc protein prepared similarly.

### 4.2. PBMC Isolation and In Vitro Cultures

Human blood samples from healthy volunteers were purchased from STEMCELL Technologies (Tukwila, WA). PBMCs were isolated by gradient centrifugation using endotoxin-free Ficoll-Paque (GE Healthcare, Chicago, IL, USA) [17]. PBMCs were placed in 24-well tissue culture plates (1 × 10^6^ cells per well) and incubated in a standard tissue incubator (Thermo Fisher Scientific) with 5% CO_2_ at 37 °C either with medium alone (cRPMI), or with two doses (100 ng and 1 μg) of soluble rhSema4A (rhSema4A-GST, Abnova), rhSema4D-Fc (R&D Systems), WT rhSema4A-Fc, M198A rhSema4A-Fc, and F223A rhSema4A-Fc proteins (GenScript). Ultra-LEAF purified mouse IgG1 served as irrelevant protein control (clone MOPC-21, BioLegend, San Diego, CA, USA). Cells were harvested 48 h later and analyzed by flow cytometry for Treg cell marker expression on live cells using the commercial Abs listed in the next section.

### 4.3. Flow Cytometry

PBMC were washed and maintained in MACS buffer (Miltenyi Biotech, Auburn, CA, USA) during the staining protocol for specific cell markers. Cells were stained first with viability marker according to the manufacturer’s instructions (Thermo Fisher Scientific LIVE/DEAD Fixable Aqua Dead Cell Stain Kit, Eugene, OR, USA) followed by incubation with an FcR block (BD Pharmingen) and, afterward, with specific Abs to the corresponding extracellular markers such as CD3-FITC and CD4-PE-Cy7 (both from BioLegend, San Diego, CA, USA), and CD25-allophycocyanin (eBioscience) were applied. To stain for intracellular markers, the Foxp3/Transcription Factor Staining Buffer Set (eBiosciences) was used as described previously with anti-Foxp3-eFluor 450 Ab (Invitrogen) [17]. The appropriate isotype control Abs were applied to separate control tubes as described and the staining profile was used to establish the negative gate [17]. After a final wash, either 10,000 cells/tube or 30,000 cells/tube out of 1 × 10^6^ cells/tube placed in MACS buffer were acquired in logarithmic scales and analyzed with the BD LSR Fortessa flow cytometer. FlowJo software was used for the live cell gating and their further analysis.

### 4.4. Direct Ligand–Receptor Interaction Assay (LRA) and Competition LRA ELISA

To analyze Sema4A- and Sema4D–Plexin B1 interaction, we employed the direct LRA and competition LRA as described previously [53,54]. Briefly, Immulon 2HB microtiter plates were coated with either rhSema4A-GST or -Fc or rhSema4D-Fc in duplicates in varying doses, starting from 2 μg/mL to 0.015 μg/mL overnight at 4 °C in 0.1M sodium carbonate-bicarbonate buffer. For receptors, the starting concentration of rhPlexin B1-His or rhNRP2-Fc was 8–10 μg/mL. In some experiments, HIS-Select nickel-coated microplates (Sigma) were used to capture Plexin B1-His (10 μg/mL). The same range of concentrations was used for proteins in solutions. ELISA was performed as described [53,54] and the amount of bound ligand was measured spectrophotometrically on the SPECTROStar Nano microplate reader (BMG Labtech) at 450 nm. Data were analyzed using a nonlinear regression analysis with GraphPrizm 8 software using the following equation:A = Amax/1 + Kd/(L)
where A represents the absorbance of the oxidized substrate, which is assumed to be proportional to the amount of ligand bound, Amax is the absorbance at saturation, (L) is the molar concentration of the ligand, and Kd is the equilibrium dissociation constant. The receptor–ligand dissociation constant (KD) was used as a measure of the receptor–ligand binding affinity.

### 4.5. BPPS/SIPRIS Analysis

BPPS [32,33,34] takes, as input, a typically very large multiple sequence alignment (MSA) consisting of members of a protein superfamily. It partitions the aligned sequences into hierarchically arranged subgroups based on the residues that most distinguish each subgroup from other, closely related subgroups. It does this using a Markov chain Monte Carlo sampling optimization strategy that stochastically moves sequences between hierarchically arranged partitions, each defined by an evolving characteristic pattern. It generates as output: (1) hierarchically arranged subgroups, along with a list of the pattern residues for each subgroup; (2) corresponding rich-text-format ‘contrast alignments’ highlighting representative sequences from each subgroup, as shown and described in Appendix A for Sema4A; and (3) PyMOL scripts to visualize the locations of BPPS-defined pattern residues within available structures. The statistical and algorithmic details regarding BPPS and evaluations of BPPS performance have been detailed in previous studies [32,33,34].

As previously described, SIPRIS [34] takes, as input, BPPS-defined residue patterns and structural coordinates for proteins from corresponding subgroups. It identifies the statistically most significant network of pattern residues embedded within a structurally defined cluster. SIPRIS computes the statistical significance as a *p*-value based on a statistical approach, termed Initial Cluster Analysis (ICA) [34,55].

### 4.6. Sequences and Structures Used in the Analysis

We used MAPGAPS (Multiply-Aligned Profiles for Global Alignment of Protein Sequences) [56] with the manually curated NCBI CDD (Conserved Domain Database) hierarchical MSA cd09295 as the query to identify and multiply align Sema domain sequences from the NCBI and translated EST (Expressed Sequence Tags) databases [57]. Due to memory limitations, we first split these databases into smaller files containing no more than 250,000 sequences each. After running MAPGAPS on each of these files separately, the resulting MSAs were merged into a single MSA. Next, we removed sequence fragments (i.e., those with >25% deletions) and all but one sequence among those sharing ≥98% identity, yielding an MSA of 4005 Sema domains. Open reading frames with a minimum of 100 residues and pdb identifiers corresponding to Sema4D (1OLZ) and Plexin B1 (3OL2; Uniprot O43157) structures [30,39] were also included in the analysis. Molecular images of 3D structures were created using the PyMOL Molecular Graphics System (Version 2.0; Schrödinger, LLC, Cambridge, MA, USA).

### 4.7. Statistics

Unpaired parametric two-tailed t test with Gaussian distribution and ordinary one-way ANOVA were used in GraphPad Prizm 8 (San Diego, CA, USA) software to calculate the statistical significance between experimental measurements.

## Figures and Tables

**Figure 1 ijms-23-03024-f001:**
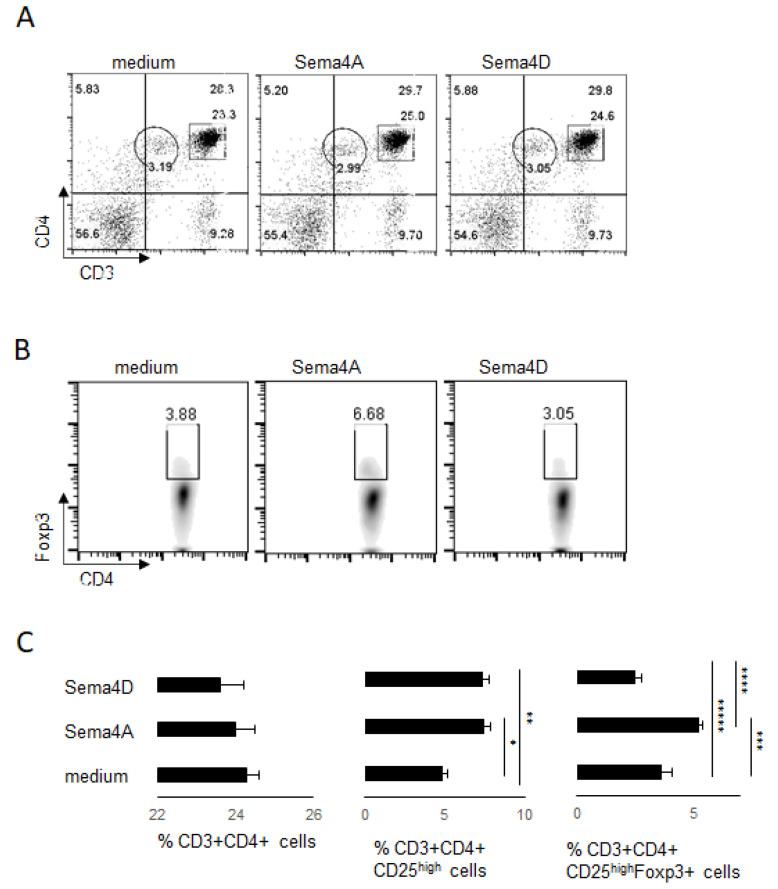
Human Sema4 proteins affect Treg cells in PBMC cultures. Human PBMC were cultured with or without 100 ng of external rhSema4A or rhSema4D for 48 h. After harvest, the numbers of CD4+ T cells (panels (**A**,**C**)) and Treg cells (panels (**B**,**C**)) were assessed by flow cytometry, using specific Abs to corresponding cell surface and intracellular molecules. CD3+CD4+ cells shown on upper panel dot plots (**A**) were further selected to evaluate the relative number of Treg cells (**B**) in each in vitro experimental setting. The Treg cells were gated on CD4+Foxp3+ density plots. (**C**) The graphical representation of data with corresponding statistics. * *p* < 0.013, cultures with medium vs. rhSema4A, ** *p* < 0.012, medium vs. rhSema4D, *** *p* = 0.012, medium vs. rhSema4A, **** *p* = 0.001, rhSema4A vs. rhSema4D, ***** *p* = 0.040, medium vs. rhSema4D. Data are representative from three independent experiments.

**Figure 2 ijms-23-03024-f002:**
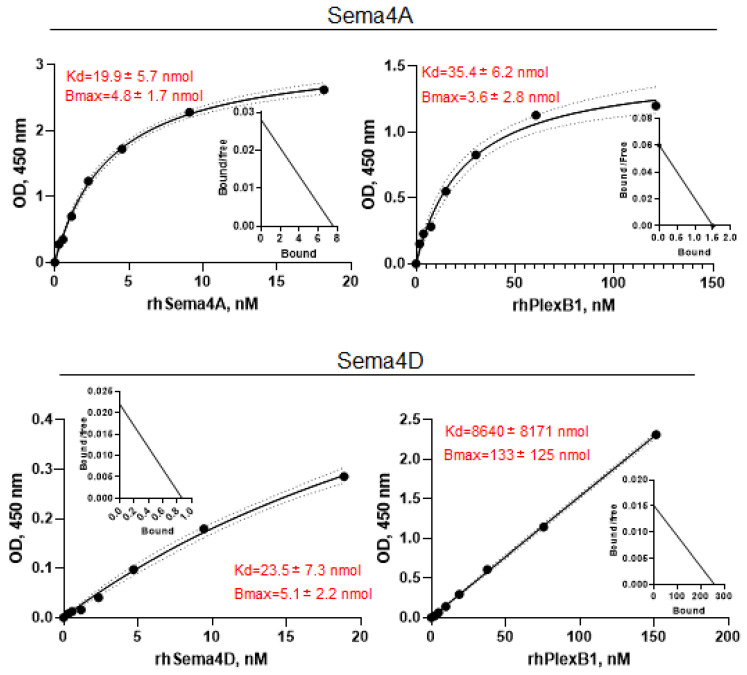
Sema4A binds Plexin B1 with higher affinity than Sema4D. Analysis of Sema4A-Plexin B1 and Sema4D–Plexin B1 interaction by the direct ligand–receptor binding ELISAs with respective Scatchard plots shown as inserts. Either Plexin B1 (8–10 μg/mL) or Sema4 (2 μg/mL) were immobilized on the Immulon 2 plates overnight and corresponding ligands or receptor were added in duplicates in varying doses, starting from 2 μg/mL for Sema4 and 10 μg/mL for Plexin B1. The amount of bound ligand was measured spectrophotometrically based on the HRP-labeled Abs to corresponding protein tags (Fc, GST, or His). Calculations of the equilibrium dissociation constants were performed utilizing GraphPrizm 8.0. software. For the KD value determination, the data was fitted to the ‘One site-specific binding with Hill slope’. Data are shown in mean ± SEM from three independent ELISA experiments. The curves demonstrate a direct 1:1 ligand–receptor binding and are representative of three independent assays.

**Figure 3 ijms-23-03024-f003:**
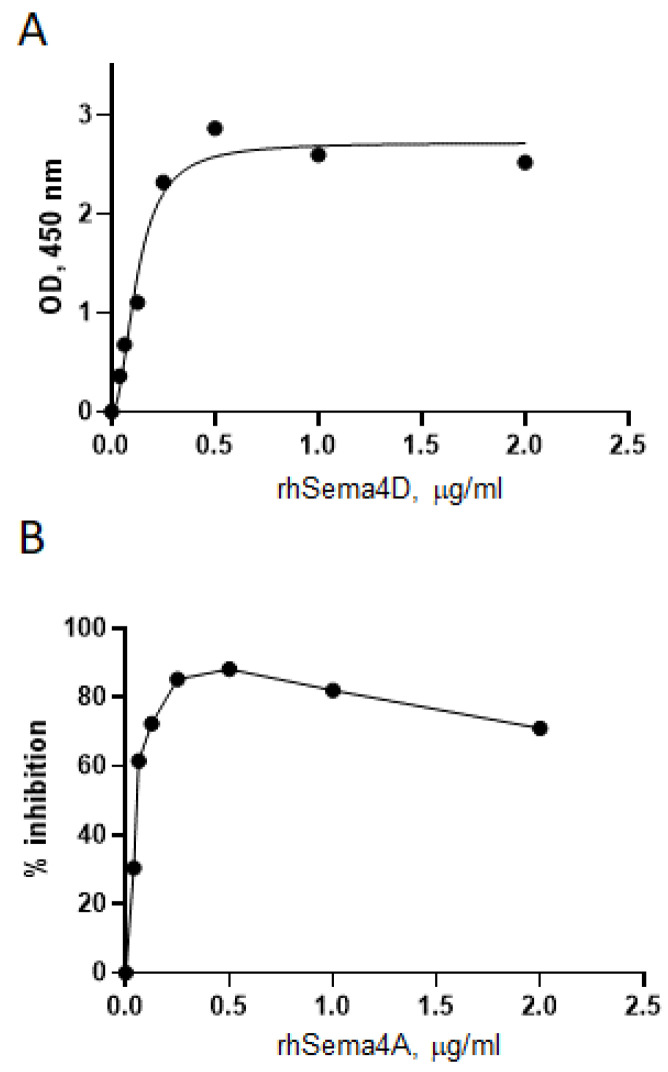
Sema4A and Sema4D compete for Plexin B1 binding in the in vitro ELISA assay. The HIS-Select nickel-coated microplates were covered with 10 μg/mL of Plexin B1 with His tag in Tris Buffer overnight. Either rhSema4A-GST or -Fc or rhSema4D-Fc in duplicates were applied to the plates in varying doses starting from 2 μg/mL. (**A**) rhSema4D binding to Plexin B1 in the direct LRA ELISA. (**B**) Inhibition of 2 μg/mL Sema4D binding to Plexin B1 by increasing concentrations of Sema4A, starting at 2 μg/mL dose. The OD for Sema4A+Sema4D was compared to the corresponding OD for Sema4D alone in the wells covered with Plexin B1 protein.

**Figure 4 ijms-23-03024-f004:**
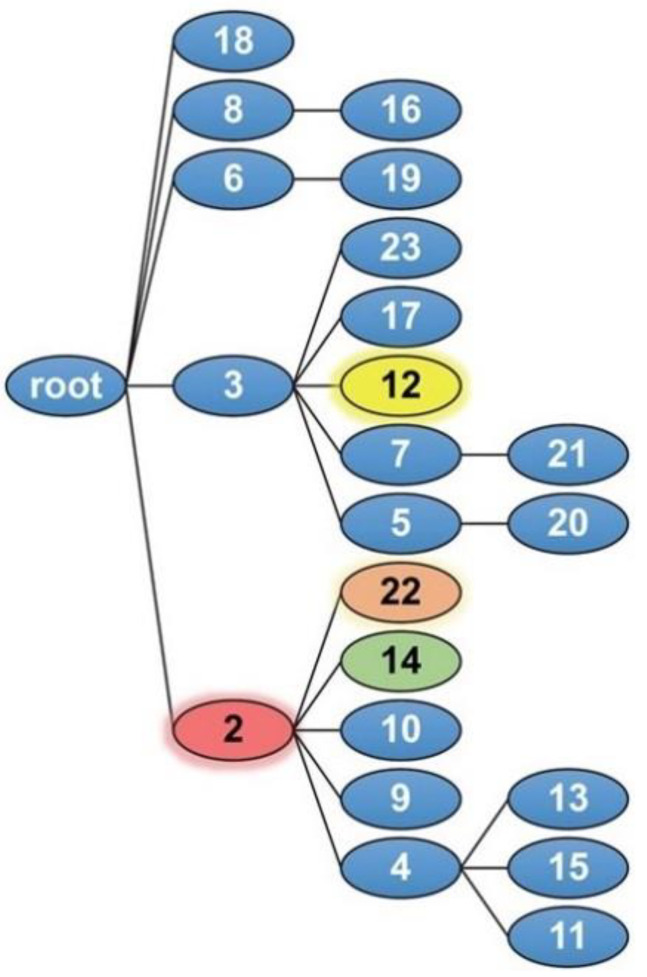
BPPS defined Sema domain hierarchy. Sema4A, Sema4D, and PlexinB1 correspond to nodes 14, 22, and 12, respectively.

**Figure 5 ijms-23-03024-f005:**
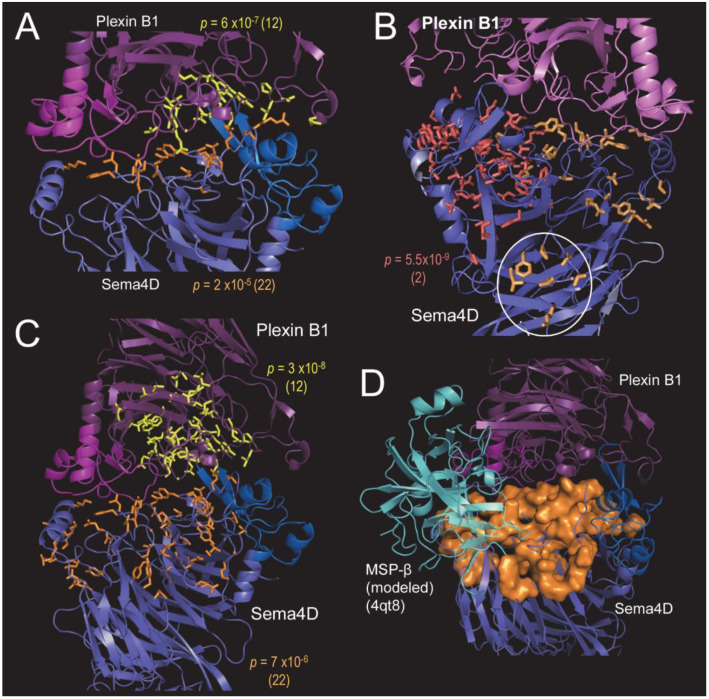
BPPS-SIPRIS analyses of the Sema4D–PlexinB1 complex (pdb_id: 3ol2). (**A**) BPPS-SIPRIS analysis identifying Sema4D- and PlexinB1-specific residues significantly clustered at the Sema4D–PlexinB1 interface; *p*-values and, in parentheses, the corresponding node IDs are indicated. Shown in orange are Sema4D residues contacting Plexin B1 and in yellow Plexin B1 residues contacting Sema4D. (**B**) Sema4D family- and subfamily-specific residue clusters identified by BPPS-SIPRIS (residue sidechains shown in red and orange, respectively). Two subfamily clusters were found: adjacent to the Plexin B1 interface and another at the Sema4D homodimeric interface (circled). (**C**) BPPS-SIPRIS analysis identifying Sema4D- and PlexinB1-specific residues significantly clustered within each subunit. (**D**) The Sema4D-specific residues in panel C (orange space-fill) are relative to a modeled interaction with MSP-β subunit. Bound MSP-β was modeled by structurally superimposing, over Plexin B1, the Ron Sema domain + MSP-β complex (pdb_id: 4qt8). Although merely hypothetical, this suggests that the larger Sema4D surface in panel C may facilitate binding of other cellular components.

**Figure 6 ijms-23-03024-f006:**
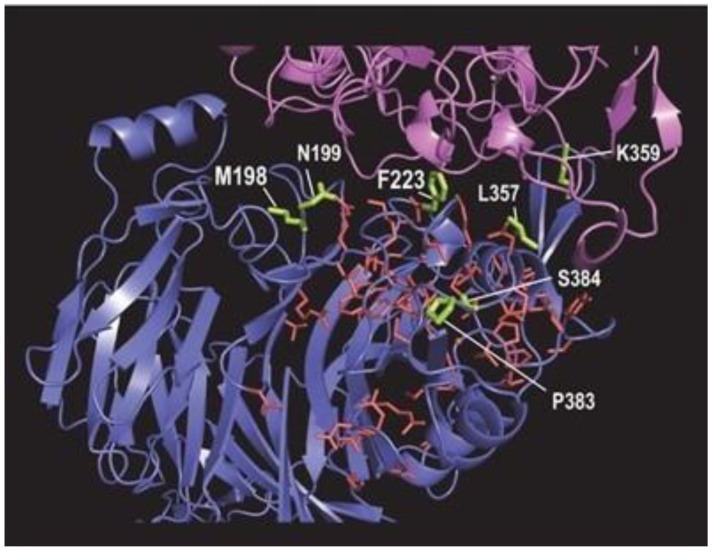
Modeling of Sema4A subfamily-specific residues (lime-colored sidechains) based on homology to the Sema4D structure. The sidechains of node 2 family residues (within Sema4D) are shown in red. The backbone of Plexin B1 is shown in pink and of Sema4D in blue.

**Figure 7 ijms-23-03024-f007:**
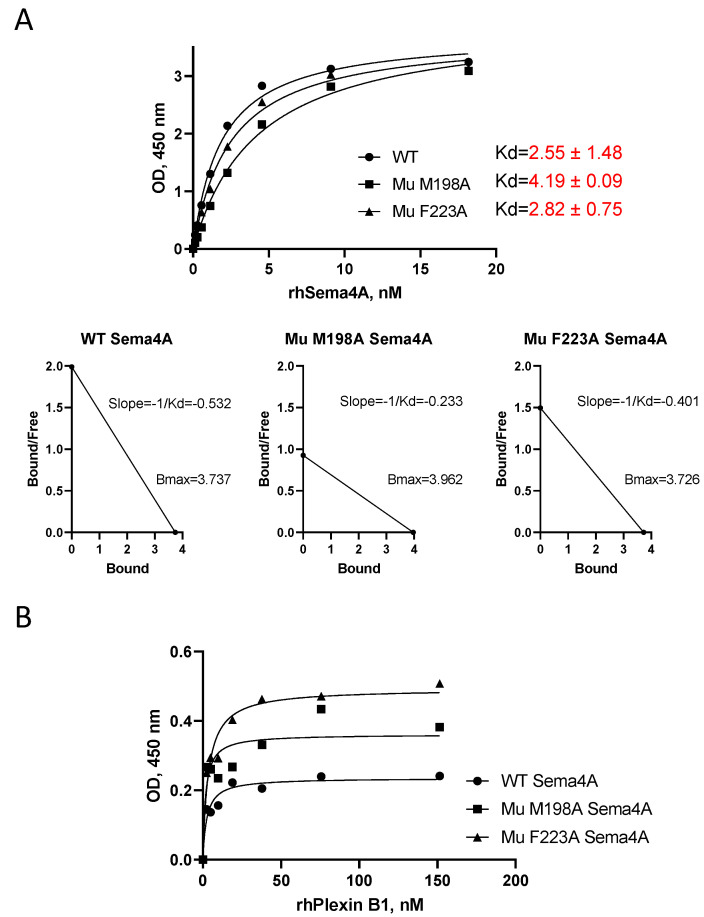
WT and Mutant (Mu) Sema4A proteins bind Plexin B1 with similar affinity. Direct (**A**) and reverse (**B**) LRA ELISAs for WT and Mu Sema4A protein binding to Plexin B1 were performed as described in Materials and Methods, analyzed in GraphPrizm 8.0 software, and corresponding Kd determination was performed as described for Figure 2. Kd are shown in mean ± SEM from three independent ELISA experiments, whereas slopes are from one representative experiment.

**Figure 8 ijms-23-03024-f008:**
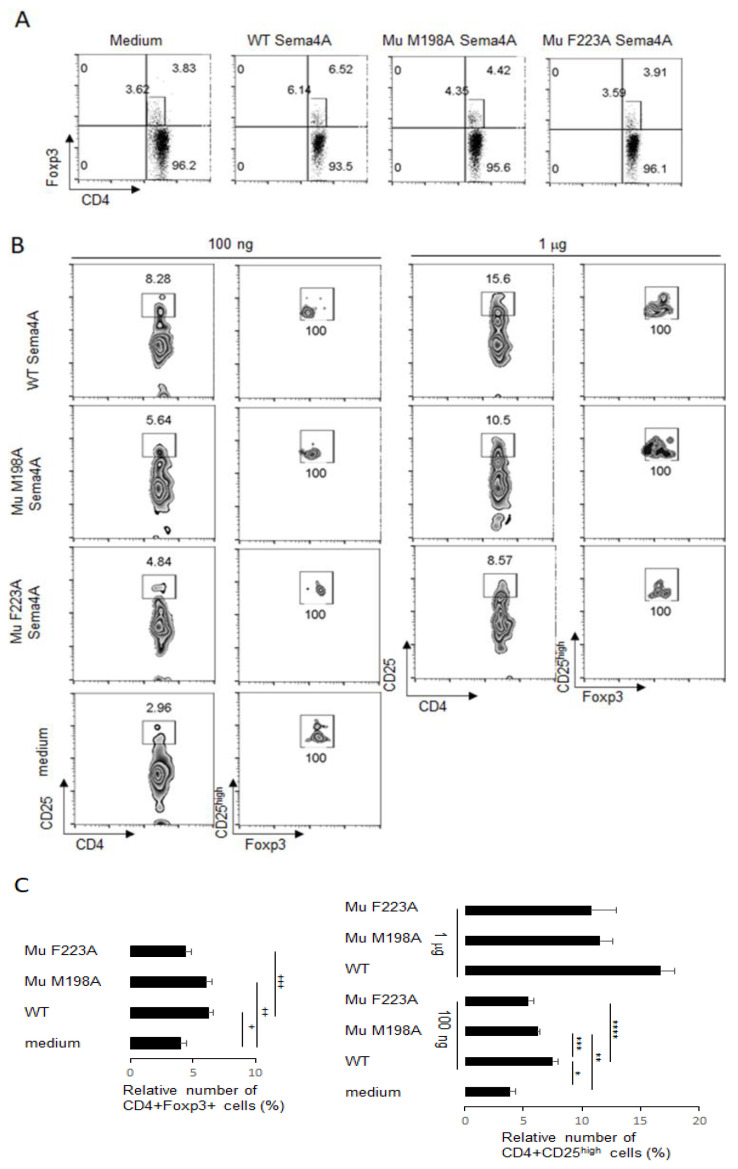
Mutant Sema4A molecules are less potent in the stabilization of human peripheral Treg cells in vitro as compared to WT recombinant Sema4A protein. Human PBMC were cultured with or without 100 ng of external WT Sema4A or Mu Sema4A for 48 h. After harvest, the numbers of Treg cells (panels (**A**,**C**)) were assessed by flow cytometry, using specific Abs to corresponding cell surface and intracellular molecules. The Treg cells were gated on CD4 + Foxp3+ dot plots. The relative number of CD3 + CD4 + CD25^high^ cells (panels (**B**,**C**)) for two distinct concentrations of individual recombinant protein in each in vitro experimental setting. Data are representative from two independent experiments. ^+^
*p* < 0.012, medium vs. WT Sema4A; ^++^
*p* < 0.02, medium vs. Mu M198A Sema4A; ^+++^
*p* < 0.038, WT vs. Mu F223A Sema4A; * *p* < 0.002, med vs. WT Sema4A, ** *p* < 0.054, med vs. Mu M198A, *** *p* < 0.041, WT vs. Mu F198A Sema4A, **** *p* < 0.016, WT vs. F223A Sema4A.

## Data Availability

The datasets generated and analyzed during the current study are available from the corresponding authors on reasonable request.

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
