# Peer review of "Identifying Function Determining Residues in Neuroimmune Semaphorin 4A"

_ijms, 2022, doi:10.3390/ijms23063024_

Round 1

Reviewer 1 Report

In this manuscript from Chapoval et al., the authors determine which aminoacids of semaphorin 4A (Sema4A) are involved in the interaction with Plexin B1. The authors report that this interaction induces a T regulatory cell  polarization by means of flow cytometry. Opposite to this finding, the authors find that semaphorin 4B elicits the contrary effect. Both semaphorins were shown to have a competitive binding to Plexin B1.

While the identification of the aminoacids implicated in the Sema4A - Plexin B1 interaction looks correct to me, I have several crucial concerns about how the authors have interpreted their results about T regulatory cell polarization in this study:

Major

  • First, the induction of FOXP3+ T regulatory cells by Sema4A is weak. In Figure 1B, the authors show a 2.5% increase in relation with the control condition. Although the authors indicate that this increase is statistically significant, this does not mean that such increase could be relevant at a biological level. In this figure, there is no a clear population of CD4+ FOXP3+ cells, but rather some disperse dots close to the CD4+ FOXP3- quadrant. The gating here seems to be crucial, and a slight variation in the gating could generate totally opposite results to what the authors are showing. 
  • Second, the flow cytometry data shown in Figure 7 looks like a convicing polarization (a clear population can be seen in the drawn box). Also, the mutations in Sema4A reduce the number of CD4+ CD25+ cells by 5% or 7%. However, it was published by Triplett et al. in 2012 that humans have a set of memory T cell characterized for being CD25+ FOXP3-. Therefore, CD25 per se can't be used as a marker for T regulatory cells. In this same Figure, once again, the changes induced by Sema4A or mutated Sema4A in the FOXP3+ population are weak in comparison to the CD25+ polarization. 

Minor

  • About the binding experiments, the authors claim that the experiment is representative of three independent experiments but only dot is shown per condition (Figures 2, 3, 7). What's the meaning of this dot? Is it a single experiment? Does it correspond to the average of three experiments? Authors should show all the data obtained from their three independent experiments. 
  • Statistical analysis: the authors indicate in Materials and Methods that parametric statistical tests were performed. If the flow cytometry experiments were performed from two independent experiments, as indicated in Figure 1, the statistical analysis is wrong. The application of parametric statistical test requires a variable with a normal and homoscedastic distribution obtained from several independent experiments, ideally higher than 6 to have a representative result. With an n = 2, there is not enough power to applicate parametric methods to calculate statistics. This explains why the statistics shown in this paper are heavily significant for such small differences.
  • The quality of presentation is bad. Some figures show good resolution while others show bad quality, there is a huge inconsistency in the image quality that makes some figures hard to interpret, especially in Figure 5. 

Author Response

Reviewer #1:

In this manuscript from Chapoval et al., the authors determine which amino acids of semaphorin 4A (Sema4A) are involved in the interaction with Plexin B1. The authors report that this interaction induces a T regulatory cell polarization by means of flow cytometry. Opposite to this finding, the authors find that semaphorin 4B elicits the contrary effect. Both semaphorins were shown to have a competitive binding to Plexin B1.

While the identification of the amino acids implicated in the Sema4A - Plexin B1 interaction looks correct to me, I have several crucial concerns about how the authors have interpreted their results about T regulatory cell polarization in this study:

Major

  1. First, the induction of FOXP3+ T regulatory cells by Sema4A is weak. In Figure 1B, the authors show a 2.5% increase in relation with the control condition. Although the authors indicate that this increase is statistically significant, this does not mean that such increase could be relevant at a biological level. In this figure, there is no a clear population of CD4+ FOXP3+ cells, but rather some disperse dots close to the CD4+ FOXP3- quadrant. The gating here seems to be crucial, and a slight variation in the gating could generate totally opposite results to what the authors are showing. 

Response: In Figure 1B we show that Sema4A exposure of human PBMC almost doubled the number of Treg cells in cultures with an increase from 3.96 +/- 0.5% in cultures with medium alone to 6.18 +/- 0.58% in cultures with rhSema4A. This increase is biologically significant as a low number of Treg cells is able to suppress Teffector cell function (1:32 dilution used in the in vitro suppression assays, please see: Dowling et al., Front Immunol. 2018 Oct 30;9:2461. doi: 10.3389/fimmu.2018.02461). Treg cells represent approximately 1–4% of peripheral blood mononuclear cells (PBMCs) (Miltenyi Biotech website). We reported previously: a. increased relative number of CD4+Foxp3+ cells among PBMCs from 3.9 ± 0.6% in cultures with medium to 6.2 ± 0.1% in cultures with 100 ng of rSema4A; b. rSema4A potentiates Treg cells in the in vitro CD4+ T cell cultures; c. Sema4A is as effective as TGF-β in induction of new Treg cells in CD4+CD25− T cell cultures (Chapoval et al., Immunohorizons. 2019 Feb;3(2):71-87). In addition to that, the reviewer can notice in supplemental Figure S1 our flow cytometry gating strategy which clearly shows that these Treg cells selected on CD4-Foxp3 positive cells are also CD25-high. A previous study from our laboratory has shown that doubling of Treg cell numbers in vivo has a profound effect on asthmatic phenotype (see: Dorsey et al., J Immunol. 2013 Aug 15;191(4):1517-28. This is now included into Discussion section of the revised manuscript). For the reviewer’s convenience, we show in the revised manuscript text the absolute numbers of Treg cells for each stimulus plus control. 

  1. Second, the flow cytometry data shown in Figure 7 looks like a convincing polarization (a clear population can be seen in the drawn box). Also, the mutations in Sema4A reduce the number of CD4+ CD25+ cells by 5% or 7%. However, it was published by Triplett et al. in 2012 that humans have a set of memory T cell characterized for being CD25+ FOXP3-. Therefore, CD25 per secan't be used as a marker for T regulatory cells. In this same Figure, once again, the changes induced by Sema4A or mutated Sema4A in the FOXP3+ population are weak in comparison to the CD25+ polarization. 

Response: In Figure 8A we show the same gating strategy as in Figure 1B whereas on panel B we show the relative numbers of CD25high cells among CD3+CD4+ cells. We agree that CD25 positivity per se can’t be used as a marker for Treg cells. We have added another panel to this Figure to show that these gated CD25high T cells are FoxP3 + Tregs as has been reported by us (reference #17) and others (reviewed in Sakaguchi et al., Foxp3+ CD25+ CD4+ natural regulatory T cells in dominant self-tolerance and autoimmune disease. Immunol Rev. 2006 Aug;212:8-27). Please note that we always carefully define the positive gates by comparing the specificity of Foxp3 staining using isotype control Ab as defined in our ImmunHorizons publication.

Minor

  1. About the binding experiments, the authors claim that the experiment is representative of three independent experiments but only dot is shown per condition (Figures 2, 3, 7). What's the meaning of this dot? Is it a single experiment? Does it correspond to the average of three experiments? Authors should show all the data obtained from their three independent experiments. 

Response: Indeed, we performed three independent binding experiments with technical replicates in each experiment. The range of OD values varied from experiment to experiment. Thus, we have calculated the binding affinity from each independent experiment, and then averaged these values and show the mean +/- SEM in revised Figure 2. We now include this information in the Materials and Methods section of the revised manuscript.

  1. Statistical analysis: the authors indicate in Materials and Methods that parametric statistical tests were performed. If the flow cytometry experiments were performed from two independent experiments, as indicated in Figure 1, the statistical analysis is wrong. The application of parametric statistical test requires a variable with a normal and homoscedastic distribution obtained from several independent experiments, ideally higher than 6 to have a representative result. With an n = 2, there is not enough power to applicate parametric methods to calculate statistics. This explains why the statistics shown in this paper are heavily significant for such small differences.

Response: Indeed, we performed three independent in vitro experiments, but each included 6 measurements (n=6) for each value. We have clarified this in the Methods section and the Figure Legend. In addition, we consulted with a statistician and the Graph Pad Prism technical guide.  Thus, we believe our statistical tests are appropriate.

  1. The quality of presentation is bad. Some figures show good resolution while others show bad quality, there is a huge inconsistency in the image quality that makes some figures hard to interpret, especially in Figure 5. 

Response: We apologize that all figures did not reproduce at high quality in the original submission. We have taken appropriate measures to ensure that figures provided in this resubmission are in tiff or jpeg format to improve their overall quality.

All changes to the manuscript are noted in red font.

Reviewer 2 Report

The manuscript proposed by Dr. Chapoval et al. is well written and very clear, and the message is clearly presented. 

check at line 366 when defining ROR-gt beacuse the "gt" is missing but there are spaces between the following word.

Author Response

Reviewer #2:

  1. The manuscript proposed by Dr. Chapoval et al. is well written and very clear, and the message is clearly presented. check at line 366 when defining ROR-gt beacuse the "gt" is missing but there are spaces between the following word.

Response: We thank Reviewer #2 for his favorable review of our manuscript. The mistake on line 366 has been corrected.

Reviewer 3 Report

In this work, Chapoval and collaborators studied the effects of Semaphorin 4A wild type and several mutants of this protein on Treg cells and on Plexin B1interactions. Although the results can be potentially interesting, this work has several weaknesses, which must be addressed:

  1. The structure of manuscript is not right, and it difficult the reading and understanding (see below comments).
  2. Introduction is very extensive and diverts the focus of attention away from the main objective. It describes in depth several previous works, but not focus on how semaphorin could be used in immunotherapeutics for asthma or for other allergic diseases. Moreover, the main objective of the work is vague and not well defined.
  3. Why authors used PBMCs from healthy donors and not from asthmatic patients? I believe that is strongly necessary to perform these experiments with cells from asthmatic patients.
  4. Statistical one-tailed tests have been performed. I suggest to perform them with two-tail.
  5. How many cells authors acquired in flow cytometry for evaluating Treg??? Moreover, graphics of flow cytometry (figures and supplemental figures) do not show the scale and units of fluorescence.
  6. Results must reflect the results obtained in your work and only compared with previous studies when it is strongly necessary. In several subsections of your results, you describe previous studies diverts your results from the focus of attention. Several parts of your results must be addressed in discussion section.

Author Response

Reviewer #3:

In this work, Chapoval and collaborators studied the effects of Semaphorin 4A wild type and several mutants of this protein on Treg cells and on Plexin B1interactions. Although the results can be potentially interesting, this work has several weaknesses, which must be addressed:

  1. The structure of manuscript is not right, and it difficult the reading and understanding (see below comments).

Response: The revised manuscript has been edited to improve its structure and logical flow.

  1. Introduction is very extensive and diverts the focus of attention away from the main objective. It describes in depth several previous works, but not focus on how semaphorin could be used in immunotherapeutics for asthma or for other allergic diseases. Moreover, the main objective of the work is vague and not well defined.

Response: The Introduction section has been shortened and re-focused as suggested by the reviewer. The main objective is clearly stated in the Abstract and Introduction section.

  1. Why authors used PBMCs from healthy donors and not from asthmatic patients? I believe that is strongly necessary to perform these experiments with cells from asthmatic patients.

Response: The goal of this study was to demonstrate that certain structural changes in the Sema4A molecule could affect its functional effect on Treg cells. The assessment of cells from asthmatic patients is beyond the scope of this current research but will be definitely considered in future experimental work.

  1. Statistical one-tailed tests have been performed. I suggest to perform them with two-tail.

Response: As suggested, we have now performed two-tailed tests. The revised manuscript was modified accordingly (Materials & Methods section and Results section, including the statistical data for Figures).

  1. How many cells authors acquired in flow cytometry for evaluating Treg??? Moreover, graphics of flow cytometry (figures and supplemental figures) do not show the scale and units of fluorescence.

Response: For evaluating Treg, we acquired from 10,000 to 30,000 cells/tube/experiment. We included into the Materials and Methods section information on the number of acquired cells and the fact that scales were logarithmic. We included into our response to Reviewer #1 details on the Treg cells gating strategy with logarithmic scales shown as well as isotype control Ab staining which served as a reference for Foxp3+ cells. 

  1. Results must reflect the results obtained in your work and only compared with previous studies when it is strongly necessary. In several subsections of your results, you describe previous studies diverts your results from the focus of attention. Several parts of your results must be addressed in discussion section.

Response: We redirected several parts of our results to the Discussion section as suggested.

Round 2

Reviewer 3 Report

Thank you for your accurate responses. Authors have answered all my previous questions. However, I have several minor comments:

Results: When p-value is written, I believe that you refer to exact p-value. Thus, you can write "p < 0.05, p < 0.01, p < 0.001", etc.; or "p = 0.04", for example. However, it is not usual to write p < 0.04 (line 114), p < 0.015 (line 115). Please, review this comment in the whole result section and in figure legends.

Materials and methods: you must include the headquarters location of each manufacturer, at least, the first time that you name it.

Page 17, Line 562: Please, replace "GraphPad Prizm 8" by "GraphPad Prism 8 (San Diego, CA, USA).

Author Response

Response:

Results. The mentioned p-values have been corrected according to the reviewer’s comments. For example, p<0.04 which equaled p=0.0398 has been changed to p=0.040, p<0.0011 changed to p=0.001 and p<0.015 to p=0.012, all mentioned on lines 114 and 115, correspondingly, of the Results. The corresponding changes were made in Figure 1 legends.

Materials and Methods. The headquarters location of each manufacturer has been added.

Page 17. Requested replacement has been done.